# Occurrence and Ecological and Human Health Risk Assessment of Polycyclic Aromatic Hydrocarbons in Soils from Wuhan, Central China

**DOI:** 10.3390/ijerph15122751

**Published:** 2018-12-05

**Authors:** Tekleweini Gereslassie, Ababo Workineh, Xiaoning Liu, Xue Yan, Jun Wang

**Affiliations:** 1Key Laboratory of Aquatic Botany and Watershed Ecology, Wuhan Botanical Garden, Chinese Academy of Sciences, Wuhan 430074, China; tekle206@gmail.com (T.G.); abiyework@gmail.com (A.W.); liuxn17@wbgcas.cn (X.L.); 2Sino-Africa Joint Research Center, Chinese Academy of Sciences, Wuhan 430074, China; 3University of Chinese Academy of Sciences, Beijing 100049, China

**Keywords:** polycyclic aromatic hydrocarbons, land-use, ecological risk, potential sources, cancer risk

## Abstract

Polycyclic aromatic hydrocarbons are large groups of ubiquitous environmental pollutants composed of two or more fused aromatic rings. This study was designed to evaluate the distribution, potential sources, and ecological and cancer risks of eleven polycyclic aromatic hydrocarbons from Huangpi soils in Wuhan, central China. The soil samples for this study were taken from 0–10 cm and 10–20 cm soil depths. A modified matrix solid-phase dispersion extraction method was applied to extract analytes from the soil samples. A gas chromatograph equipped with a flame ionization detector was used to determine the concentrations of the compounds. The sum mean concentrations of the polycyclic aromatic hydrocarbons were 138.93 and 154.99 µg kg^−1^ in the 0–10 cm and 10–20 cm soil depths, respectively. Benzo[*a*]pyrene and fluorene were the most abundant compounds in the 0–10 cm and 10–20 cm soil depths, respectively. The quantitative values of the pyrogenic index, total index, and diagnostic ratio used in this study showed that the polycyclic aromatic hydrocarbons have a pyrogenic origin. The negligible and maximum permissible concentrations values for naphthalene, acenaphthylene, acenaphthene, phenanthrene, anthracene, pyrene, benz[*a*]anthracene, and benzo[*a*]pyrene indicated a moderate ecological risk. The incremental lifetime cancer risk values for adults and children showed a low and moderate cancer risk, respectively.

## 1. Introduction 

Environmental pollution has become a crosscutting issue attracting the attention of countries’ governmental, public, and scientific communities in the last few decades [1,2]. Polycyclic aromatic hydrocarbons (PAHs) are a large group of abundant, widespread hydrophobic environmental pollutants with two or more fused aromatic rings [3,4]. Polycyclic aromatic hydrocarbons are characterized by high lipid solubility, facile bioaccumulation, environmental toxicity, and persistent nature, with high melting and boiling points and low vapor pressure [5,6]. They are environmentally ubiquitous compounds found in measurable concentrations in different environmental components (water, soil, and sediments) [7]. Polycyclic aromatic hydrocarbons often originate from anthropogenic sources such as heat-induced organic matter decomposition, incomplete fossil fuel combustion, accidental petroleum spills, factory discharge, smelting, garbage incineration, coal burning, and motor vehicle emissions [4,8,9]. Polycyclic aromatic hydrocarbons can also be produced from natural sources such as forest and peat fires, biological processes, volcanic eruption, and ancient sediment erosion [10,11,12].

Polycyclic aromatic hydrocarbons end up in various environmental components; however, soil and sediments are considered as the primary steady reservoir and sinks [11,13]. Polycyclic aromatic hydrocarbons are thousands in number, but the United States Environmental Protection Agency (USEPA) classified 16 PAHs as priority carcinogenic and mutagenic compounds [14,15]. Depending on their molecular structure and number of aromatic rings, PAHs can be grouped into low-molecular weight PAHs (LMW_PAHs_), with two or three aromatic rings, and high-molecular weight PAHs (HMW_PAHs_), with four or more aromatic rings [16,17]. Low-molecular weight (LMW_PAHs_) are environmentally abundant and highly toxic compounds; however, they are relatively less persistent, have lower carcinogenicity, and are more easily degradable than high-molecular weight (HMW_PAHs_) [13,14,16]. Polycyclic aromatic hydrocarbons can induce a broad spectrum of acute and chronic health impacts such as malformation, mutagenesis, and endocrine system disruptions [3,18,19,20]. 

Researchers have affirmed that about half of China’s twenty-seven major lakes and freshwater sources are extremely contaminated with persistent organic pollutants (POPs) [21,22]. Consequently, using polluted fresh water for irrigation and untreated solid wastes as fertilizer can induce serious soil pollution problems. Besides the application of fertilizers and pesticides, urbanization and industrialization, coupled with a lack of strict environmental supervision, are among the key factors exacerbating the environmental pollution in China, and the Wuhan vicinity in particular [23,24]. The Huangpi district is one of the 13 districts located in the north of the Wuhan metropolitan. Many industrial plants (cement plants, foundries, coal gasification and coking plants) have been built in the city of Wuhan, and rapid urbanization facilities are expected to cause soil pollution [25]. Huangpi is among the districts which are expected to experience substantial environmental pollution originating from urbanization, industrialization, and urban municipal wastes. The district covers a large area of agricultural land. The dietary demand of a large portion of the population in the city of Wuhan depends on crops, vegetables, and fruits produced in the Huangpi district. Thus, the exposure of humans to pollution from PAHs is expected to be high. In view of this, monitoring and evaluating the status of PAHs in soil is crucial. Therefore, this study was designed to investigate the concentration, spatial distribution, potential sources, and ecological and human health risks of PAHs in soil from the Huangpi district, Wuhan, China.

## 2. Materials and Methods 

### 2.1. Sample Collection and Pretreatment

The Huangpi district is located between 30°52′30′′ N and 114°22′30′′ E. The district covers a total area of 2261 km^2^, of which 1267.46 hectares (56.12%) is cultivated land, 4124 hectares (18.29%) is forest land, 4.98 hectares (0.22%) is grassland settlements, and 160.07 hectares (7.09%) is comprised by industrial sites, 4.76 hectares (0.21%) by land transport, 296.50 hectares (12.3%) by water bodies, and 92.09 hectares (4.08%) by unused land [26]. The total population it hosts was 874,938 in 2010 [27] and 1.13 million in 2015 [26]. Soil samples were collected from eighteen sampling places. The collected samples belong to four different land-use types, namely: barren (BL), farmland (FL), plastic greenhouse (PGH), and paddy field (PF), sampled at the depths of 0–10 cm and 10–20 cm using a precleaned stainless steel grab sampler. Locations for each sampling point were recorded using a global positioning system (GPS). Sample 17 and sample 18 were collected from a similar location, thus one GPS reading was used for these two samples. Consequently, 17 sampling sites are shown on the map (Figure 1). 

Three soil samples were collected from each sampling site and immediately wrapped in polyethylene zip-lock bags. Then, the samples from each site were combined and lyophilized using a benchtop lab vacuum freeze-dryer at −40 °C for 48 h. All the samples were ground and passed through a 100-mesh (0.149 mm stainless steel sieve) [15] and stored in a refrigerator at −20 °C until the next extraction steps [28].

### 2.2. Chemicals and Standards 

A 2000 mg L^–1^ mixture of USEPA priority PAHs containing naphthalene (NaP), acenaphthylene (Acy), acenaphthene (Ace), fluorene (Flr), phenanthrene (Phe), anthracene (Ant), fluoranthene (Flu), pyrene (Pyr), benz[*a*]anthracene (BaA), chrysene (Chr), benzo[*b*]fluoranthene (BbF), benzo[*k*]fluoranthene (BkF), benzo[*a*]pyrene (BaP), dibenz[*a,h*]anthracene (dbA), benzo[*ghi*]perylene (BgP), indeno[*1,2,3-cd*]pyrene (InP), and pyrene (Pyr) was purchased from ANPEL laboratory technology (Shanghai Inc., Shanghai, China). Chromatographic grade dichloromethane and acetonitrile were obtained from Fisher Scientific, (Waltham, MA, USA), and Mallinckrodt Baker, Inc., USA, respectively. In addition, analytical grade reagents: carbon (C18) (SiliCycle, Inc., Quebec City, QC, Canada), anhydrous sodium sulfate (Na_2_SO_4_) and copper powder (Cu) (Sinopharm Chemical Reagent Co. Ltd., Shanghai, China). Florisil (Beijing Yizhong Chemical Plant, Beijing, China), and neutral silica gel (Qingdao Haiyang Chemical Co., Qingdao, China) were purchased from the respective companies. Pretreatment and activation of the reagents were carried out to improve the effectiveness of the extraction. Accordingly, the anhydrous sodium sulfate was baked at 450 °C for 4 h before use. The 60–100 mesh Florisil and 100–200 mesh neutral silica gel were activated by oven-drying at 150 °C for 10 h and 180 °C for 4 h, respectively. The neutral silica gel was deactivated with 3% ultrapure water before use. Unlike the other reagents, copper was activated by soaking the copper powder in a 2 N solution of HCl (Sinopharm Chemical Reagent Co. Ltd.) for 12 h, and was then washed three times with water and three times with acetone [29].

### 2.3. Sample Extraction 

The target PAHs were extracted using a modified matrix solid-phase dispersion extraction method described previously [29,30,31]. One gram of soil sample was thoroughly blended with three grams of C18 using a glass mortar and pestle for five minutes [15]. One gram of activated Na_2_SO_4_, one gram of Florisil, one gram of neutral silica gel, and one gram of copper powder were packed into a blank 10 mL syringe barrel-column with a 0.22 μm membrane filter from bottom to top. After the blended mixture had completely transferred into the syringe column using a funnel, the column was closed with a 0.22 μm membrane filter and compressed with a syringe plunger to remove the air. The sodium sulfate (Na_2_SO_4_), Florisil, and neutral silica gel were added to remove any traces of water and form a free-flowing solution, while activated copper powder was added to remove elemental sulfur [32]. Elution was carried out using 20 mL of dichloromethane by gravity flow [29]. Finally, the extracts were concentrated to 50 µL under a gentle high-purity nitrogen (N) stream, and 50 µL of acetonitrile was added as a solvent keeper [28]. 

### 2.4. Instrumental Analysis 

A gas chromatograph (GC Hewlett-Packard HP-5890; Los Angeles, CA, USA) equipped with flame ionization detection (GC-FID) was used to determine the concentration of PAHs [33]. One microliter of each sample extract was automatically injected into an HP-5 capillary column (30 m × 320 μm × 0.25 μm). The gas chromatograph column temperature was programmed to begin at 40 °C (hold for 1 min) and was ramped up to 280 °C at a rate of 10 °C/min (hold for 2 min), while the detector was operated at 310 °C. To increase the quality of the experiment, the glassware used was thoroughly cleaned with distilled water, baked in a muffle furnace (Zhengzhou Protech Furnace Co. Ltd., Zhengzhou, China) at 450 °C for 4 h, and rinsed with hexane (Sinopharm Chemical Reagent Co. Ltd) before use. Besides, the gas chromatograph was calibrated using standard solutions of 0.1 ppm, 0.5 ppm, 1 ppm, 2 ppm, 5 ppm, and 10 ppm dissolved in acetonitrile. The calibration curve for the compounds was linear with an average correlation coefficient value (*R*^2^) of 0.991, which implied that the detection potential of the instrument during the experiment was good. Interference of samples during the instrumental analysis was checked by including one blank for every ten samples. No PAHs were detected in the blank samples. The average recoveries in this study varied from 80 to 115%. The minimum concentration of compounds that could be obtained from the noise (limit of detection) and the lowest concentration of the analytes in the samples (limit of quantifications) were determined as a function of three and ten times the standard deviation of the blank, respectively [20]. The limits of detection and quantification for the soil samples ranged from 0.008 to 0.168 µg kg^−1^ and 0.025 to 0.560 µg kg^−1^, respectively. 

In addition to PAHs, three soil properties were determined following the standard methods recommended by the Chinese Society of Soil Science [30]. The total organic carbon (TOC) contents of the soil were analyzed with the Shimadzu TOC analyzer (Shimadzu, Kyoto, Japan) (SSM-5000A). Soil pH was determined using a pH meter (PHS-3C; Leici, Ningbo, China) in a suspension of 1:5 soil-to-water ratio [30,34], while soil moisture content (MC) was determined by drying a known weight of soil in an oven-dryer at 105 °C for 24 h until a constant weight was recorded. 

### 2.5. Ecological Toxicity and Risk Assessment 

Evaluation of the ecological toxicity of the PAH compounds in the soils was carried out by comparing the risk quotient (RQ) of PAHs under investigation and their corresponding environmental quality values. There is insufficient toxicological data for PAHs in agricultural soil [35]. The environmental quality concentration values negligible concentration (NCs and maximum permissible concentration (MPCs) for Phe, Ant, Flu, BaA, Chr, and BaP were taken from [36]. Researchers have agreed that PAHs with the same toxicity equivalence factor (TEF) have similar human and ecological health impacts [28,37]. Therefore, the environmental quality values of congeners with the same TEF were used to calculate the environmental risk quotient of Nap, Acy, Ace, Flr, and Pyr. The individual and total environmental risk quotients were determined using the following equations:
(1)$$\mathrm{RQ}=\frac{\mathrm{CPAHs}}{\mathrm{CQV}}$$
(2)$$\mathrm{RQNCs}=\frac{\mathrm{CPAHs}}{\mathrm{CQV}(\mathrm{NCs})}$$
(3)$$\mathrm{RQMPCs}=\frac{\mathrm{CPAHs}}{\mathrm{CQV}(\mathrm{MPCs})}$$
(4)$$\mathrm{RQ}\sum \mathrm{PAHs}=\sum_{n=1}^{11}\mathrm{RQi}\;(\mathrm{RQi}\geq 1)$$
(5)$$\mathrm{RQ}\sum \mathrm{PAHs}(\mathrm{NCs})=\sum_{n=1}^{11}\mathrm{RQi}(\mathrm{NCs})\;(\mathrm{RQi}(\mathrm{NCs})\geq 1)$$
(6)$$\mathrm{RQ}\sum \mathrm{PAHs}(\mathrm{NCs})=\sum_{n=1}^{11}\mathrm{RQi}(\mathrm{MPCs})\;(\mathrm{RQi}(\mathrm{MPCs})\geq 1)$$
where CPAHs is the concentration of certain PAHs in soil, CQV is the corresponding quality values of certain PAHs, NCs and MPCs are, respectively, the negligible and maximum permissible concentrations of PAHs in soil, RQ is the risk quotient, CQVNCs are the quality values of the NCs of PAHs in soil, and CQVMPCs are the quality values of MPCs in soil. 

The total summed environmental risk RQ∑PAHs was calculated by considering the individual RQMPCs and RQMPCs ≥1. The environmental risk levels for individual and total PAHs are summarized in Table 1. In addition to the risk quotients, the carcinogenic risk of individual PAHs was calculated from the concentrations of each PAHs and their corresponding toxicity equivalency factor [38]. 

Human health impacts of persistent organic pollutants can arise through dermal contact, ingestion, and inhalation. Human risk assessments of PAHs were evaluated by using the equations adopted from previous work [11,39]. The toxicity equivalency quotient (TEQ) is an important tool to estimate the relative toxicity of individual PAH fractions compared to benzo[*a*]pyrene [39]. The TEQ and incremental lifetime cancer risk (ILCR) of each PAH through direct ingestion, dermal contact, and inhalation for adults and children were calculated using Equations (7–10):
(7)$$\mathrm{TEQ}=\sum_{n=1}^{n}\mathrm{Ci}*\mathrm{TEFi}$$
(8)$$\mathrm{ILCRSingestion}=\frac{\mathrm{CS}\times (\mathrm{CSFIngestion}\times \sqrt[{3}]{\frac{\mathrm{BW}}{70}}\times \mathrm{IRsoil}\times \mathrm{EF}\times \mathrm{ED})}{\mathrm{BW}\times \mathrm{AT}\times \mathrm{cf}}$$
(9)$$\mathrm{ILCRSdermal}=\frac{\mathrm{CS}\times (\mathrm{CSFDermal}\times \sqrt[{3}]{\frac{\mathrm{BW}}{70}}\times \mathrm{SA}\times \mathrm{AF}\times \mathrm{ABS}\times \mathrm{EF}\times \mathrm{ED})}{\mathrm{BW}\times \mathrm{AT}\times \mathrm{cf}}$$
(10)$$\mathrm{ILCRSinhalation}=\frac{\mathrm{CS}\times (\mathrm{CSFinhalation}\times \sqrt[{3}]{\frac{\mathrm{BW}}{70}}\times \mathrm{IRair}\times \mathrm{EF}\times \mathrm{ED})}{\mathrm{BW}\times \mathrm{AT}\times \mathrm{PEF}\times \mathrm{cf}}$$
where TEQ is the toxicity equivalence quotient, TEF is the toxicity equivalence factor, CS is the PAH concentration of soils (µg kg^−1^), CSF is the carcinogenic slope factor (µg kg^−1^ day^−1^)^−1^, CSF was based on the cancer-causing ability of BaP, and the CSFingestion, CSFdermal, and CSFinhalation of BaP were 7.3, 25, and 3.85 (µg kg^−1^ day^−1^)^−1^, respectively [11]. BW is body weight (70 kg), AT is average life span (70 years), EF is exposure frequency (350 days year^−1^), ED is the exposure duration (30 years), IRsoil is the soil intake rate (100 mg day^−1^), IRair is the inhalation rate (20 m^3^ day^−1^), SA is the dermal surface exposure (5000 cm^2^ day^−1^), cf is the conversion factor (10^6^), AF is the dermal adherence factor (10 mg cm^−2^), ABS is the dermal adsorption fraction unitless (0.1), and PEF is the soil dust produce factor (1.32 × 10^9^ m^3^ kg^−1^). The total risks were the sum of risks of the ILCRs in terms of direct ingestion, dermal contact, and inhalation. For children, BW (15 kg), EF (180 days), ED (6 year), IRair (10 m^3^ day^−1^), SA (2800 cm^2^ day^−1^), and AF (0.2 kg cm^−2^) were assumed [9].

### 2.6. Statistical Analysis

Microsoft Excel 2010 (Microsoft; Redmond, WA, USA) was used to organize and arrange the data. All the descriptive statistics: minimum, maximum, average, and other computations were conducted using SPSS statistics version 20 (IBM; Amonk, NY, USA). One-way analysis of variance (ANOVA) was used to check if there was a statistical difference in concentrations between individual PAHs, soil depth, and land-use types. Pearson’s correlation analysis was also used to determine the connection between PAHs and soil properties. 

## 3. Result and Discussion

### 3.1. Concentrations of Polycyclic Aromatic Hydrocarbons in Soil

The average, range, standard deviation, and total concentrations of the 11 PAHs: Acy, Ace, Flr, Phe, Ant, Flu, Pyr, BaA, Chr, NaP, and BaP are illustrated in Table 2. The total concentrations of the PAHs ranged from 6.22 to 376.93 µg kg^−1^ (∑mean = 138.93 µg kg^−1^) and 10.36 to 450.59 µg kg^−1^ (∑mean = 154.99 µg kg^−1^) in the 0–10 cm and 10–20 cm soil depths, respectively. Compared to results of other studies, the summed mean concentration of the current study was markedly lower than the 16,380 µg kg^−1^ reported from roadside soil in India [40]; the 2936.1 to 5282.3 µg kg^−1^ from core sediments of Lake Hongfeng, southwest China [41]; 294 to 12741 µg kg^−1^ from the coastal region of Macao, China [42]; 1174.33 µg kg^−1^ from the Majia River and 914.40 µg kg^−1^ from the Tuhai–Majia River system, China [43]; 2052.6 µg kg^−1^ from urban soil of Xi’an, northwest China [9]; 428.41 µg kg^−1^ from bank soils in Danjiangkou Reservoir, China [28]; and the mean concentration of 209 µg kg^−1^ in soil of the Songhua River basin, China [44]. However, the results were comparable with the 79.7 to 473 µg kg^−1^ reported in bank soils from the three Gorges, China [45]; 198 µg kg^−1^ in soil from Midway Atoll, the north Pacific Ocean [11]; and 36.9–378 µg kg^−1^ from bank soils of Luan River, China [37]. The relatively lower concentration of PAHs in this study might be due to less use of untreated wastewater and limited industrial discharges released into the soil. BaP, Ace, and Flr were the dominant PAHs in both the 0–10 cm and 10–20 cm soil depths. Concurrently, the total average concentration of LMW_PAHs_ was 73.02 µg kg^−1^ in the surface and 91.46 µg kg^−1^ in the subsequent layer, while the concentrations of HMW_PAHs_ were 65.91 and 63.53 µg kg^−1^ in the two soil depths, respectively. The variation in concentration of PAHs with soil depth is due to a higher photochemical and biological degradation rate in the surface soil [46]. One-way analysis of variance (ANOVA) revealed that there was a significant variation in the concentration of individual PAHs (*p* < 0.05). 

The spatial distribution of PAHs across different land-use types and sampling sites in the study area was evaluated by comparing the obtained average, range, and total concentrations of PAHs. It can be observed from Table 3 that relatively higher concentrations of PAHs in the 0–10 cm soil depth were recorded for S3, S7, S8, S9, S10, S11, S12, and S15, whereas for S1, S2, S4, S5, S6, S13, S14, S16, S17, and S18, they were higher at 10–20 cm depth. Overall, 75% of the samples that were from farmland and barren land showed higher concentrations of PAHs in the topsoil layer. None of the samples collected from plastic greenhouses showed higher concentrations in the surface layer. The spreadsheet of the individual PAHs for all sampling points and the two soil depths is displayed in the Appendix A (see Table A1).

The mean concentrations of PAHs in all samples ranged from 2.03 µg kg^−1^ (S4) to 30.04 µg kg^−1^ (S11) and 1.31 µg kg^−1^ (S8) to 35.89 µg kg^−1^ (S5) in the top and subsequent soil layers, respectively. The average concentration of PAHs in the surface soil layers of the four land-use types were in the order of BL>PF>FL>PGH, with concentrations of 79.45, 54.54, 50.73, and 40.71 µg kg^−1^, respectively. In contrast, the summed average concentrations of PAHs in the subsequent soil layer were higher in PGH (87.63 µg kg^−1^), followed by PF (68.53 µg kg^−1^), BL (57.83 µg kg^−1^), and FL (39.62 µg kg^−1^). The higher concentration of PAHs in the 10–20 cm soil layers of PGH could be due to less exposure to environmental factors such as temperature, runoff, sunlight, and the affinity of PAHs to bind with stable soil particles. On the other hand, farmland samples showed a lower level of PAHs than the paddy field and barren land. This might be due to the influence of continuous tillage and other human interference accelerating the photo-oxidation, volatilization, and diagenesis in farmland. The summed mean concentrations of PAHs from both depths of the four land-use types were highest in BL (68.64 µg kg^−1^), followed by PGH (64.17 µg kg^−1^), PF (61.54 µg kg^−1^), and FL (45.18 µg kg^−1^). However, the one-way analysis of variance showed no statistically significant difference (*p* < 0.05) in the concentration of PAHs among the land-use types and the two soil depths.

According to the literature, the environmental distribution and availability of PAHs depend on their potential sources and pathways [6,14], tendency to bind with organic matter [17], soil stability, and microorganisms [6]. Some researchers also stated that due to the high level of TOC in the surface soil, the concentration of PAHs often show a gradual increment from deeper to surface segments of soil [17,41]. Therefore, the relatively high concentration of PAHs in the surface soil layer of areas of barren land might be associated with the availability of TOC, accumulation of pollutants from petroleum products in flowing water, high soil stability, surface-laying water, and the soil microorganisms current. Likewise, additions from rainwater and lower rates of volatilization due to the availability of high plant remains might contribute to the enrichment of PAHs in the top layers of barren land.

In addition to the above points, the authors of [40] stated that the concentration of PAHs varies with location, season, meteorological conditions, photochemical processes, and distance from road networks. Thus, the lower concentration of PAHs in the surface layer of PGH in this study is due to the limited additions from surface runoff, rain washout, dry atmospheric particle deposition, garbage incineration, road traffic, and asphalt pavements. The comparison of the obtained concentration of PAHs with those of other studies has been summarized as follows (Table 4).

According to the authors of [52], there are four proposed threshold PAH contamination levels for soil: not contaminated (<200 µg kg^−1^), weakly contaminated (200–600 µg kg^−1^), contaminated (600–1000 µg kg^−1^), and heavily contaminated (>1000 µg kg^−1^). According to these classifications [52], the majority of the sampling sites were grouped under the “not contaminated” classification. Only three samples, from Zhulinyuan (S7), Leqianwan (S11), and Bomogang (S12) in the 0–10 cm group, and five samples from Tangjiawan (S1), Zhujiashan (S5), Hanjiafan (S13), Wanjiatian (S14), and Dujiatian (S17) in the 10–20 cm group were grouped under “weakly contaminated” (Table 3). The total average concentrations of 138.72 µg kg^−1^ in the 0–10 cm group and 153.27 µg kg^−1^ in the 10–20 cm group indicated that the soil is not contaminated with PAHs. The low level of soil contamination in this study might be due to a limited direct discharge of PAHs containing urban wastes and products of traffic combustion [14]. 

### 3.2. Relationship Among Polycyclic Aromatic Hydrocarbons and Selected Soil Properties 

Many researchers have included soil properties as key parameters during the determination of PAHs in soils and other environmental components. Thus, soil moisture content, soil total organic carbon content, and soil pH were the soil properties determined in this study. The results obtained ranged from 0.63 to 2.71%, from 4.35 to 47.70%, and from 4.12 to 8.07 for TOC, MC, and pH, respectively. The Pearson’s correlation coefficient value obtained in this study showed a significant positive correlation among individual PAHs, except for Nap and BaA (R = −0.053). Acy showed a strong significant correlation with all PAHs except BaA (*p* ≤ 0.01). While BaA showed a significant positive correlation only with Pyr and Flu, this implied that BaA has a common source with Pyr and Flu. Generally, the strong positive correlation among PAHs implied that they have a common origin. The detailed values of the relationships among PAHs and selected soil properties are displayed in the Appendix A (see Table A2)**.**


Pearson’s correlation coefficient results indicated weak negative and positive correlations between PAHs and the selected soil properties. All the PAHs investigated in this study, except BaP (R = 0.191) and Ant (R = 0.137), showed a weak negative correlation with TOC. pH exhibited a negative correlation with all PAHs except Nap (R = 0.011), Chr (R = 0.079), and Ant (R = 0.007). Only Ant (R = 0.139), BaA (R = 0.099), and Flr (R = 0.022) showed a weak positive correlation with soil moisture content. Ace (R = −0.328), Acy (R = −0.317), and Nap (R = −0.291) exhibited relatively higher negative correlations with TOC, pH, and MC, respectively. Similarly, the authors of [53] indicated that all PAHs except (Ace, Flu, and Chr) showed a negative and no correlation with TOC and pH, respectively. Likewise, a weak positive correlation between TOC and the concentration of PAHs was reported by the authors of [54] (R = 0.10), [55] (R = 0.054), [33] (R = 0.57), and (R = 0.39) [29]. Other previous research [44] concluded that PAHs have no significant correlation with TOC in soil.

### 3.3. Possible Sources of Polycyclic Aromatic Hydrocarbons 

Researchers commonly determine the distribution and potential sources of PAHs in the environment using diagnostic ratios such as Phe/Ant, Flu/Pyr, BaA/Chr, Flu/Flu+Pyr, and Ant/Ant+Phe [56]. Researchers have also suggested that the pyrogenic index (PI), i.e., the ratio of LMW to HMW or vice versa, is applicable to determine the potential sources of PAHs [57]. LMW_PAHs_ are dominated by a homologous series of five petrogenic alkylated PAHs (naphthalene, phenanthrene, dibenzothiophene, fluorene, and chrysene), while the HMW_PAHs_ are chiefly pyrogenic compounds [16]. Therefore, using the PI can better reflect the potential sources of PAHs. According to the authors of [58], applying the PI has some merits over the PAH isomers ratio for three reasons: first, any change in the ratio can truly reflect the change in LMW and HMW_PAHs_; secondly, the PI gives better accuracy with great consistency and less uncertainty; and lastly, natural weathering and biodegradation slightly alter the PI values. In addition to the diagnostic ratio and PI, researchers use a total index (TI) to identify the high-temperature (combustion) or low-temperature (petroleum) sources of PAHs. TI, which is the ratio of (Flu/Flu+Pyr)/0.4 + (Ant/Ant+Phe)/0.1 + (BaA/BaA+Chr)/0.2 as used previously [31,59], was employed in this study. A TI > 4 indicates that PAHs have originated mainly from combustion, while lower values indicate petrogenic sources [59]. For this particular study, identification of the potential sources of PAHs in soil was carried out using PAH diagnostic ratios, PI, and TI. The obtained results are presented in Table 5.

The LMW/HMW_PAHs_ ratios (PI values) for all the sampling sites were <1. The PI values in this study, which ranged between 0.18 and 0.84, were similar with the PI values of a previous study [58]. BaA/Chr < 0.50, Ant/Ant+Phe < 0.10, Phe/Ant > 1, Flu/Pyr > 1, and Flu/Flu+Pyr < 0.5 are indicators of pyrogenic PAHs [52,56,58]. Accordingly, all the diagnostic ratios: Phe/Ant (0.73 to 10.52), Flu/Pyr (0.20 to 1.06), BaA/Chr (0.20 to 9.23), BaA/BaA+Chr (0.17 to 1.0), Flu/Flu+Pyr (0.13 to 0.51), and Ant/Ant+Phe (0.09 to 0.58) indicated a pyrogenic origin of the PAHs. Results of this study were concurrent with those reported in other works [43,59,60,61], whereas the ratios for Ant/Ant+Phe and BaA/BaA+Chr were higher than the corresponding ranges (0.11–0.16) and (0.38–0.48) of the ratios reported in [41]. In addition, the TI values obtained in this study showed that all soil samples except S6, S10, and S17 were >4, which indicates that the PAHs determined in this study originated from combustion processes. Generally, the diagnostic ratios, PI, and TI values suggested that pyrogenic sources of contamination are predominant; therefore, it is possible to conclude that the PAHs in Huangpi soil have a mainly pyrogenic origin, having been emitted by industrial processes, transport, local heating sources, biomass burning, motor vehicle exhausts, heavy industrial emissions, and additions from plant remains. Sheshui River, which is a potential source of irrigation water in some of the sampling sites, such as Erpaiqu, Xinyang, Leqianwan, Tianjiaxiaowan, and Wanjiatian, might be one of the possible routes by which petrogenic PAHs are transported into the study area. 

### 3.4. Ecotoxicological and Human Health Risk Assessment 

The mean RQMPCs values of all PAHs in both depths were <1. Ace (0.219) and BaP (0.131) in the 0–10 cm, and Ace (0.194) and Pyr (0.169) 10–20 cm group exhibited higher RQMPCs. Chr (0.001) showed the lowest RQMPCs in both soil depths. The RQMPCs, RQNCs, and TEQ values of the PAHs investigated in this study are presented in Table 6. 

A RQMPCs value of >1.0 signifies that PAHs in the environment can cause substantial contamination, while RQNCs <1.0 implies that the occurrence of PAHs can pose negligible contamination [62,63]. The RQNCs values for Nap, Acy, Ace, Phe, Ant, Pyr, BaA, and BaP in the 0–10 cm soil layer, and Nap, Acy, Ace, Flr, Phe, Ant, Pyr, BaA, and BaP in the 10–20 cm soil layer were higher than 1.0, while the RQMPCs of individual PAHs were all less than 1.0, indicating that the listed PAHs pose a moderate level of ecological risk in the study site. Flu, Flr, and Chr on the top and Chr and Flu in the subsequent soil layer were risk-free PAHs. The ∑RQNCs values of 65.90 µg kg^−1^ in the 0–10 cm soil layer and 70.13 µg kg^−1^ in the 10–20 cm soil layer were less than the threshold ecological risk values (800 µg kg^−1^) used in [37]. 

The incremental lifetime cancer risk values for adults ranged as such: 1.13 × 10^−8^–3.54 × 10^−8^, 1.94 × 10^−6^–6.06 × 10^−6^, and 9.06 × 10^−13^–2.83 × 10^−12^ for ILCRingestion, ILCRdermal, and ILCRinhalation, respectively. Meanwhile, the ILCR values for children were ranged from 9.23 × 10^−7^–9.51 × 10^−6^, 1.62 × 10^−6^–2.77 × 10^−4^, and 4.78 × 10^−11^–2.31 × 10^−10^ for ingestion, dermal contact, and inhalation, respectively. According to the literature [9,11,15], ILCR ≤ 10^−6^ indicates negligible potential cancer and human health risks, while values from 10^−6^ to 10^−4^ indicate low cancer risk. Thus, the obtained cancer risk values for adults revealed that the PAHs can bring a negligible carcinogenic impact through ingestion and inhalation, whereas the ILCRdermal (1.94 × 10^−6^–6.06 × 10^−6^) for adults indicates a low potential cancer risk. The ILCRingestion (9.23 × 10^−7^–9.51 × 10^−6^) and ILCRdermal (1.62 × 10^−6^–2.77 × 10^−4^) values for children indicated low and moderate cancer risk, respectively. ILCR values for children were higher than those for adults. The ILCR results in this study were comparable to those reported in [11] in soils from the north Pacific Ocean. The ILRC values for ingestion, dermal contact, and inhalation for adults and children are displayed in the Appendix A (see Table A3). 

All PAHs in this study except BaP exhibited TEQ values of <1, suggesting that these levels of PAHs cannot induce a harmful human health impact [59,64]. The TEQ values of the eleven PAHs investigated in this study, arranged in descending order, are as follows: BaP > BaA > Chr > Ant > Ace > Flr > Pyr > Phe > Flu > Acy > Nap, in the 0–10 cm soil layer, while the TEQ values in the 10–20 cm soil layer were in the order of: BaP > BaA > Ant > Chr > Flr > Ace > Pyr > Phe > Flu > Acy > Nap. The results of this study were similar to the findings reported in [65] for a drinking water source from a large mixed-use reservoir in China. Among the PAHs investigated in this study, only BaP can induce a carcinogenic impact. Generally, the total TEQ values in the top (34.55 µg kg^−1^) and in the next layer (24.74 µg kg^−1^) were much lower than the safe soil levels recommended by the Canadian Council of Ministers of the Environment (600 µg kg^−1^) [66].

## 4. Conclusions 

Eighteen soil samples collected from different land-use types were analyzed in this study. The mean concentrations of the PAHs investigated in this study were higher in the soil layer of 10–20 cm depth than that of 0–10 cm depth. LMW_PAHs_ constituted 53% and 59% of the total PAH concentration in the surface and subsequent soil layers, respectively. BaP in the surface and Flr in the subsurface soil layer were the most highly concentrated PAHs. The obtained total mean concentrations of PAHs from both layers of different land-use types decreased in the order of BL > PGH > PF > FL. Due to fewer petrogenic sources nearby and limited application of untreated irrigation water, the concentrations of PAHs in this study were lower than the concentrations of PAHs reported from different parts of China and other countries. The Pearson’s correlation coefficient results showed strong, moderate, and relatively weak positive relationships among the investigated PAHs. However, PAHs showed mostly weak negative with infrequent weak positive correlations with soil moisture content, pH, and total organic carbon. The diagnostic ratios, PI, and TI values indicated that the PAHs detected in the study area have a pyrogenic origin. The RQNCs and RQMPCs values indicated that the majority of the PAHs investigated in this study have a moderate level of ecological risk. The TEQ values of the PAHs investigated in this study were less than the threshold ecological risk and safe levels for soil. ILCR results revealed a low potential of cancer risk for adults and a moderate risk for children posed by the PAHs in the study area. 

## Figures and Tables

**Figure 1 ijerph-15-02751-f001:**
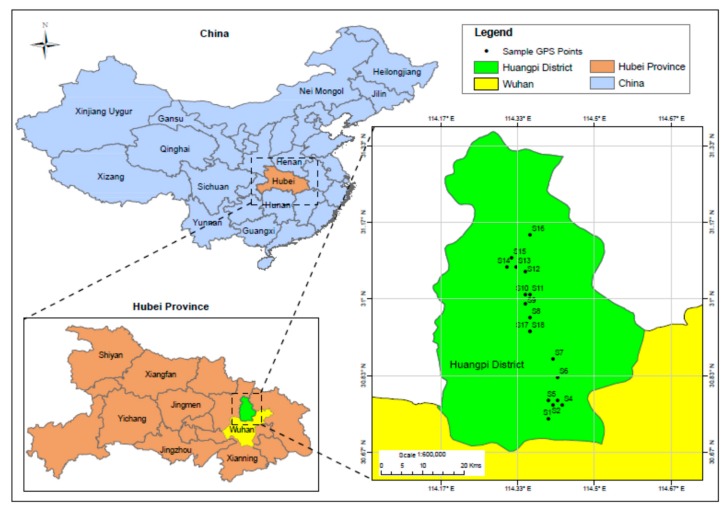
Map of the study area and sampling locations. Note: S1 (Tangjiawan), S2 (Fengdouhu) S3 (Erpaiqu), S4 (Changdi), S5 (Zhujiashan), S6 (Tujiadun), S7 (Zhulinyuan). S8 (Zhoujiawan), S9 (Lishuwan), S10 (Xinyang), S11 (Leqianwan), S12 (Bomogang), S13 (Hanjiafan), S14 (Wanjiatian), S15 (Hongguanshanxiawan), S16 (Dujiatian), S17 (Tianjiaxiaowan), S18 (Tianjiaxiaowan).

**Table 1 ijerph-15-02751-t001:** The environmental quality values negligible concentration and maximum permissible concentration of polycyclic aromatic hydrocarbons (PAHs).

Individual PAHs			∑PAHs		
Risk Grade	RQ(NCs)	RQ(MPCs)	Risk Grade	RQ∑PAHs (NCs)	RQ∑PAHs (MPCs)
Lower risk	<1	<1	Risk-free	<1	<1
Medium risk	≥1	<1	Low risk	≥1; <800	<1
High risk	≥1	≥1	Medium risk 1	≥800	<1
			Medium risk 2	<800	≥1
			High risk	≥800	≥1

RQ: risk quotient. Source: [29,37].

**Table 2 ijerph-15-02751-t002:** Concentrations, limits of detection (DL), and limits of quantification (LQ) of PAHs (µg kg^−1^) at depths of 0–10 and 10–20 cm.

PAHs	Number of Rings	DL	LQ	0–10 cm Depth			10–20 cm Depth		
				Range	Mean	SD	Range	Mean	SD
Nap	2	7.6 × 10^−3^	2.5 × 10^−2^	Nd–19.65	3.93	5.30	Nd–4.7	1.80	1.71
Acy	3	4.8 × 10^−2^	1.6 × 10^−1^	0.34–15.11	7.51	4.90	0.35–23.03	7.98	5.90
Ace	3	2.2 × 10^−2^	7.4 × 10^−2^	0. 70–55.95	26.31	16.71	2.29–44.77	23.31	12.68
Flr	3	3.3 × 10^−2^	1.1 × 10^−1^	2.69–34.04	19.79	9.89	1.69–57.18	30.56	13.80
Phe	3	42 × 10^−2^	1.3 × 10^−1^	1.35–20.58	10.60	6.37	3.49–48.22	16.41	12.49
Ant	3	1.5 × 10^−2^	5.0 × 10^−2^	Nd–19.5	4.88	5.59	0.16–33.57	11.41	9.91
**Σ** **LMW**				**5.0** **8–164** **.83**	**73.02**	**48.76**	**7.98–211.47**	**91.46**	**56.49**
Flu	4	1.9 × 10^−2^	6.3 × 10^−2^	0.05–30.98	7.74	7.80	0.60–25.97	8.31	7.38
Pyr	4	4.8 × 10^−2^	1.6 × 10^−1^	0.86–35.16	15.27	8.62	8.04–225.06	20.25	13.91
BaA	4	1.7 × 10^−1^	5.6 × 10^−1^	0.23–8.98	3.50	2.86	0.84–96.01	4.69	4.85
Chr	4	5.7 × 10^−2^	1.9 × 10^−1^	Nd–22.03	5.39	6.18	Nd–115.31	6.30	7.01
BaP	5	1.3 × 10^−1^	4.3 × 10^−1^	Nd–114.95	34.01	26.98	Nd–629.41	23.98	33.59
**ΣHMW**				**1.14–212.10**	**65.91**	**52.44**	**2.38–239.12**	**63.53**	**66.74**
**ΣPAHs**				**6.22–376.93**	**138.93**	**101.20**	**10.36–450.59**	**154.99**	**123.23**

Ace = acenaphthene, Acy = acenaphthylene, Ant = anthracene, BaA: benzo[*a*]anthracene, BaP = benzo[*a*]pyrene, Chr = chrysene, Flu = fluoranthene, Flr = fluorene, Nap = naphthalene, Phe = phenanthrene, Pyr = pyrene, Nd = Not detected, SD = Standard deviation, ∑LMW = sum of low-molecular weight PAHs, ΣHMW = sum of high-molecular weight PAHs, ΣPAHs = the sum of LMW and HMW PAHs.

**Table 3 ijerph-15-02751-t003:** Spatial distribution and concentration of PAHs in different land-use types.

Land-Use Type	Sites	Mean (0–10 cm)	Mean (10–20 cm)	SD (0–10 cm)	SD (10–20 cm)	Range (0–10 cm)	Range (10–20 cm)	Sum (0–10 cm)	Sum (10–20 cm)
**Farmland (FL)**	S3	9.31	7.27	7.07	8.19	0.06–21.13	Nd–23.91	102.17	79.77
	S12	20.80	12.04	21.17	16.52	Nd–69.95	0.17–55.76	249.43	126.62
	S15	9.71	7.72	9.50	8.77	0.32–25.07	0.22–29.31	106.61	63.47
	S16	10.91	12.59	10.78	10.25	Nd–38.65	0.01–32.27	119.74	138.23
**∑PAHs FL**		**50.73**	**39.62**	**12.13**	**10.93**	**0.38** **–** **154.8**	**0.4** **–** **141.25**	**577.95**	**408.09**
**Pooled mean and sum**		**45.18**						**493.02**	
**Paddy field (PF)**	S9	10.60	5.61	9.84	5.04	Nd–33.35	Nd–14.54	116.43	61.52
	S10	7.50	7.02	7.59	5.64	Nd–22.67	Nd–20.39	82.47	76.96
	S13	17.07	19.01	11.25	12.50	1.56–39.86	1.98–36.93	187.51	208.92
	S14	6.65	20.50	8.64	15.08	0.13–27.67	4.27–46.93	73.01	225.32
	S18	12.72	16.39	17.22	18.17	Nd–56.16	Nd–57.18	139.68	180.08
**∑PAHs PF**		**54.54**	**68.53**	**10.91**	**11.29**	**1.69–179.71**	**6.3–175.97**	**599.10**	**752.80**
**Pooled mean and sum**		**61.54**						**675.95**	
**Plastic greenhouse (PGH)**	S4	2.03	13.11	3.50	20.60	Nd–12.22	0.46–72.44	22.15	143.95
	S5	9.42	35.89	11.01	34.16	Nd–37.40	4.5–125.88	103.53	394.58
	S6	16.54	16.68	11.19	11.11	2.95–37.41	2.61–33.84	181.78	183.21
	S17	12.72	21.95	12.01	17.42	1.09–38.25	0.50–49.30	139.67	241.21
**∑PAHs PGH**		**40.71**	**87.63**	**9.43**	**20.82**	**4.04–125.28**	**8.0–281.46**	**447.13**	**962.95**
**Pooled mean and sum**		**64.17**						**705.04**	
**Barren land (BL)**	S1	3.86	18.97	2.32	12.61	0.63–7.52	1.88–43.53	42.25	208.45
	S2	8.48	17.77	6.82	13.50	0.67–20.74	1.04–39.43	92.99	195.24
	S7	23.40	14.05	17.24	14.79	4.87–57.69	Nd–42.72	257.19	154.28
	S8	13.67	1.31	16.82	1.21	Nd–55.95	Nd–3.49	150.09	14.21
	S11	30.04	5.73	30.17	6.89	2.75–114.95	4.27–46.93	330.28	62.80
**∑PAHs BL**		**79.45**	**57.83**	**14.67**	**9.8**	**8.9** **2–256** **.85**	**7.1** **9–176** **.1**	**872.80**	**634.98**
**Pooled mean and sum**		**68.64**						**753.89**	

**Table 4 ijerph-15-02751-t004:** Comparison of the concentrations of PAHs (µg kg^−1^) with those obtained at other locations.

Places	Environmental Component	Number of PAHs determined	Range	Mean	References
Huangpi, China	Soil (0–10 cm)	11	6.22–376.93	138.93	This study
	Soil (10–20 cm)		10.36–450.59	154.99	
Hanfeng Lake, Three Gorges, China	Bank soils	15	79.7–473		[45]
Luan River, China	Sediments	16	20.9–287	115.3	[37]
	Bank soils		36.9–378	141.4	
Soltan Abad River, Iran	Sediments	16	180.3–36	264.55	[47]
Open-pit coal mine, soil, China	0–20 cm	16	2160–335	11,940	[31]
	20–50 cm	16	230–369	9210	
	50–100 cm	16	60–36,460	6590	
Czech Republic	Agricultural soil	16	861–10,840	5527	[48]
	Forest soils		7657–79,385	25,510	
Forest of the São Paulo State, Brazil	Cunha	16		180	[49]
	PEFI			818	
Agricultural soils, South Korea	Paddy soil	16	38.3–1057	209	[50]
	Upland soil		23.3–2834	270	
Three Gorges Dam region, China	Agricultural soils	16	277.79–3217.20	1023.48	[15]
East Lake, China	Sediments	16	10.9–2478.10	685.8	[31]
Dongjiang River, China	Sediments	16	100–3400	880	[51]
North Pacific Ocean	Surface soils	16	3.55–3200	198	[11]

PEFI: Parque Estadual das Fontes do Ipiranga, São Paulo.

**Table 5 ijerph-15-02751-t005:** The diagnostic ratios, pyrogenic index (PI), and total index (TI) of the investigated PAHs.

Samples	Phe/Ant	BaA/Chr	BaA/BaA+Chr	Flu/Pyr	Ant/Ant+Phe	Flu/Flu+Pyr	∑other PAHs	∑LMW PAH	PI (LMW/HMW)	TI
1	3.88	1.67	0.63	0.35	0.20	0.26	31.77	10.71	0.34	5.82
2	0.86	0.37	0.27	0.39	0.54	0.28	62.39	30.84	0.49	7.44
3	1.56	4.89	0.83	0.35	0.39	0.26	82.17	20.23	0.25	8.70
4	0.92	0.41	0.29	0.15	0.52	0.13	18.15	4.23	0.23	7.01
5	1.02	0.57	0.36	0.42	0.50	0.30	70.44	33.15	0.47	7.53
6	4.14	0.26	0.20	0.35	0.19	0.26	115.64	66.35	0.57	3.62
7	8.39	0.63	0.39	0.89	0.11	0.47	182.95	74.46	0.41	4.18
8	5.49	9.23	0.90	0.40	0.15	0.28	103.53	46.80	0.45	6.76
9	1.01	0.69	0.41	0.60	0.50	0.37	87.52	29.13	0.33	7.96
10	3.23	0.24	0.19	0.21	0.24	0.17	44.81	37.67	0.84	3.76
11	0.90	0.20	0.17	0.38	0.53	0.28	236.33	94.18	0.40	6.79
12	5.14	1.08	0.52	0.33	0.16	0.25	189.32	60.33	0.32	4.84
13	1.19	0.66	0.40	0.64	0.46	0.39	128.66	59.08	0.46	7.52
14	0.91	0.85	0.46	0.28	0.52	0.22	59.46	13.68	0.23	8.08
15	1.42	0.61	0.38	0.20	0.41	0.17	82.37	24.47	0.30	6.45
16	0.73	6.59	0.87	1.06	0.58	0.51	84.48	35.49	0.42	11.42
17	10.52	0.69	0.41	0.62	0.09	0.38	87.16	52.74	0.61	3.86
18	0.79	4.84	1.00	0.26	0.56	0.20	118.78	21.14	0.18	11.10
**Sum**	**52.09**	**34.49**	**8.68**	**7.87**	**6.5**	**5.19**	**1785.9**	**714.67**	**7.29**	**122.8**

**Table 6 ijerph-15-02751-t006:** Computed Risk Quotient (negligible concentration and maximum permissible concentration) and Toxicity Equivalency Quotient Values of PAHs.

PAHs	PAH Quality Values			0–10 cm				10–20 cm			
	NCs µg kg^−1^	MPCs µg kg^−1^	TEF	AvCPAHs µg kg^−1^	RQ(NCs)	RQ(MPCs)	TEQ	AvCPAHs µg kg^−1^	RQ(NCs)	RQ(MPCs)	TEQ
Nap	1.4	140	0.001	3.93	2.81	0.028	0.004	1.80	1.29	0.013	0.002
Acy	1.2	120	0.001	7.51	6.26	0.063	0.008	7.98	6.65	0.066	0.008
Ace	1.2	120	0.001	26.31	21.92	0.219	0.026	23.31	19.42	0.194	0.023
Flr	2.6	260	0.001	19.79	0.76	0.008	0.020	30.56	1.18	0.012	0.031
Phe	5.1	510	0.001	10.59	2.08	0.021	0.011	16.41	3.22	0.032	0.016
Ant	1.2	120	0.01	4.88	4.07	0.041	0.049	11.41	9.51	0.095	0.114
Flu	2.6	260	0.001	7.74	0.30	0.003	0.008	8.31	0.32	0.003	0.008
Pyr	1.2	120	0.001	15.27	12.73	0.127	0.015	20.24	16.87	0.169	0.020
BaA	2.5	250	0.1	3.51	1.40	0.014	0.351	4.69	1.88	0.019	0.469
Chr	107	10,700	0.01	5.39	0.50	0.001	0.054	6.30	0.59	0.001	0.063
BaP	2.6	260	1	34.01	13.08	0.131	34.01	23.98	9.22	0.092	23.98
**Sum**	**128.6**	**12** **,** **8** **60**	**1.127**	**138.92**	**65.90**	**0.655**	**34.55**	**154.99**	**70.13**	**0.696**	**24.74**

AvCPAHs is the average concentrations of PAHs, RQMPCs is the maximum permissible level risk quotient values, RQNCs is the negligible risk quotient values, and TEQ is the toxicity equivalency quotient.

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
