# Peer review of "Occurrence and Ecological and Human Health Risk Assessment of Polycyclic Aromatic Hydrocarbons in Soils from Wuhan, Central China"

_ijerph, 2018, doi:10.3390/ijerph15122751_

Round 1
Reviewer 1 Report
Comments to the manuscript ijerph-381224. Title: "Occurrence, Ecological and Human Health Risk Assessment of Polycyclic Aromatic Hydrocarbons in Soils from Wuhan, Central China" for the International Journal of Environmental Research and Public Health.
This manuscript is about the analysis of a common family of pollutants, PAHs in soils from Central China. It is an interesting work to have levels of this family on different types of soils according their usages. However, I have some comments that I below describe.
1. Previous considerations
Please, revise English in all the text in general. For instance, line 27 or 74 (PAHs concentration).
Besides, take care of some technical words:
· Line 22. Remove brackets for ‘PAHs’.
· Line 26 and 130. The common abbreviation for the Electron Capture Detector is ‘ECD’.
· In general text and table 2 footnote, some of the PAHs are usually written with letters between brackets in italics: benz[a]anthracene, benz[b]fluoranthene, benz[k]fluoranthene, benz[a]pyrene, dibenz[a,h]anthracene, benz[g,h,i]perylene, and indeno[1,2,3-cd]. I leave this to editor consideration.
· Line 106. Na2SO4 (numbers in subscript type).
· Line 110. In my opinion, Florisil must be written with capital letter.
· All the text. Change unit Kg into kg.
· In my opinion, all number must show ‘,’ to identify thousands as for instance in line 218 (16,380).
· In table 4, please, remove one ‘,’ for Huangpi (China) in the first raw of the first column.
· Authors have employed the word ‘medium’ to talk about what I think is ‘environment’ along the text. To me, it is not a right word.
· Table 7 (Appendix). Please, add a bold line between results for S18 and Averages in 10-20 cm depth as in the 0-10 cm depth.
· Table 8 (Appendix). Check the font size of the first value in the ILCRingestion. In this same table 8, I think the second column corresponds to ILCRDermal instead of inhalation.
2. Materials and methods paragraph.
· Please, is the supplier for C18, anhydrous sodium sulfate and Cu the same company than for dichloromethane and acetonitrile? If not, please, add it, with country of origin. Add the same information about quality, trademark and country for Florisil, silica gel (line 110), water ultra-pure (line 112) and acetone (line 114).
· Authors talk about 18 sample sites, but in the map appearing in figure 1, I only can find 17. Maybe I am wrong, but I have tried it in several occasions.
· Talking about the percentage of any type of uses of lands, in line 85, I understand that water area means 12.3% out of the total area.
· Please, change GC machine (line 137) into Gas Chromatograph or GC instrument. Remove ‘machine’.
· I consider that blanks are essential in any type of analytical study. However, in my opinion, just one blank for every ten samples is few for this study.
· My main objection in this study is the use of GC with ECD. Although it is frequent to find studies with GC-FID or GC-MS, it is known that ECD is very sensitive for halogenated compounds (mainly organochlorinated pesticides or PCBs), however it is not common to use it for detection of compounds with no significant electronegativity and limited selectivity, rather poor precision of results cannot be avoided. I would understand this election if a derivatization would have been carried out. That is what, for example, Liu et al., used in J. Environ. Sci. 2007, 19, 1-11 to detect nitro-PAHs.
· Please, unify the style for text under the equations 1-6 and 7-10 (bold, size, etc.).
3. Results
· In this section, lines 212-213, there is a description of average, range, standard deviation and total concentrations of 11 PAHs. I am surprised to find NaP included in this set, because it was not cited out in the Chemicals and Standards paragraph on Material and Methods (lines 99-104). Besides, in the footnote of table 2, the abbreviation for pyrene is Pyr, but in line 104 it is written as LnP. Please, check all names all along the manuscript.
· Could you add information or comment the sentence in line 243: ‘This is highly associated with application of lindane and dicofol in farmlands’? I don’t understand if you are trying to say that high concentration of PAHs in topsoil layer also happen for dicofol and lindane? I cannot find a relationship between PAHs and those pesticides in this case. Please, explain this item to me.
· Line 264. According data from table 3, mean concentration for total PAH in paddy fields is 61.54 µg kg-1.
· Authors are using numbers to indentify sample sites (S1-S18). Please, add the corresponding number when talk about them using their names (lines 286-289).
· In table 5 two diagnostic ratios are not right written: Ph/An (it should be Phe/Ant) and Flu/Py (Flu/Pyr). In this last case, I am not sure if it must be Flu or Flr. Ther is not an accordance between the table and the text.
· Talking about Ecotoxicological and human Health Risk Assessment, in line 348, I think that the right mean value of RQ(MPCs) is 0.219 in the 0-10 cm layer.
· Table 6. Please, to simplify and better understand, add a footnote explaining what is the meaning of every head of the columns.
· I think there is a mistake in the series of TEQ values in line 372-373. According results, I think that the right order is BaP (34.01) < BaA (0.351) > Chr (0.054) > Ant (0.049) > Ace (0.026) > Flr (0.020) > Pyr (0.015) > Phe (0.011) > Flu (0.008) > Acy (0.008) > Nap (0.004).
· Please, check if the total TEQ of this study (line 376) is right. According data from table 6, I think it is 34.55. Please, verify it.
4. References
· References 10, 12, 27, 28, and 36 are not found in the text. I have been unable to find them.
· All citations along the text should be placed in squared brackets instead of normal brackets.
· Please, follow the instructions for authors to adapt all references.
Author Response
REVIWER 1
· Please, revise English in all the text in general. For instance, line 27 or 74 (PAHs concentration).
Response: We have considered the comments and made a revision
· Line 22. Remove brackets for ‘PAHs’
Response: “PAHs” removed from (line 22) and mentioned in introduction (line 44).
· Line 26 and 130. The common abbreviation for the Electron Capture Detector is ‘ECD’.
Response: The instrument used for analysis was written wrongly. The instrument used was Gas chromatography flame ionization detector (GC-FID) (line 148-149)
· In general text and table 2 footnote, some of the PAHs are usually written with letters between brackets in italics: benz[a]anthracene, benz[b]fluoranthene, benz[k]fluoranthene, benz[a]pyrene, dibenzo[a,h]anthracene, benz[g,h,i]perylene, and indeno[1,2,3-cd]. I leave this to editor consideration.
Response: Corrections included in (line 116-118) and table 2
· Line 106. Na2SO4 (numbers in subscript type).
Response: Na2SO4 changed by Na2SO4 and mentioned (line 123, 137, 142)
· Line 110. In my opinion, Florisil must be written with capital letter.
Response: florisil changed by “Florisil” (line 124, 128, 137, 142)
· All the text. Change unit Kg into kg.
Response: Kg has replaced by “kg”
· In my opinion, all number must show ‘,’ to identify thousands as for instance in line 218 (16,380).
Response: comma added
· In table 4, please, remove one ‘,’ for Huangpi (China) in the first raw of the first column.
Response: comma removed (page 16)
· Authors have employed the word ‘medium’ to talk about what I think is ‘environment’ along the text. To me, it is not a right word.
Response: “Medium” was used to express environmental components such soil and sediments etc…). it has changed by “sample type” (page 16)
· Table 7 (Appendix). Please, add a bold line between results for S18 and Averages in 10-20 cm depth as in the 0-10 cm depth.
Response: Bold line added (page 24)
· Table 8 (Appendix). Check the font size of the first value in the ILCRingestion. In this same table 8, I think the second column corresponds to ILCRDermal instead of inhalation.
Response: 4th column in table 8 ILCRinhalation replaced by ILCRdermal (page 25)
· Please, is the supplier for C18, anhydrous sodium sulfate and Cu the same company than for dichloromethane and acetonitrile? If not, please, add it, with country of origin. Add the same information about quality, trademark and country for Florisil, silica gel (line 110), water ultra-pure (line 112) and acetone (line 114).
Response: Producer companies, trade marks for the chemicals and reagents have included (page 7)
· Authors talk about 18 sample sites, but in the map appearing in figure 1, I only can find 17. Maybe I am wrong, but I have tried it in several occasions.
Response: Samples 17 and 18 were collected from one place, but different land-use type. It has explained in the methodology part (line 99-101).
· Please, change GC machine (line 137) into Gas Chromatograph or GC instrument. Remove ‘machine’.
Response: GC machine has replaced by “Gas chromatograph”
· My main objection in this study is the use of GC with ECD. Although it is frequent to find studies with GC-FID or GC-MS, it is known that ECD is very sensitive for halogenated compounds (mainly organochlorinated pesticides or PCBs), however it is not common to use it for detection of compounds with no significant electronegativity and limited selectivity, rather poor precision of results cannot be avoided. I would understand this election if a derivatisation would have been carried out.
Response: I am so sorry and I misspell the word. In fact, we use GC-FID in this study. I have correct GC-ECD to GC-FID. The full name of the instrument we used is written in (148-149)
· Please, unify the style for text under the equations 1-6 and 7-10 (bold, size, etc.).
Response: The format for all the equations has verified
· In this section, lines 212-213, there is a description of average, range, standard deviation and total concentrations of 11 PAHs. I am surprised to find NaP included in this set, because it was not cited out in the Chemicals and Standards paragraph on Material and Methods (lines 99-104). Besides, in the footnote of table 2, the abbreviation for pyrene is Pyr, but in line 104 it is written as LnP. Please, check all names all along the manuscript.
Response: NaP was missed from the list of standards, now included (line 114). The “LnP” has replaced by “Pyr” (line 116)
· Could you add information or comment the sentence in line 243: ‘This is highly associated with application of lindane and dicofol in farmlands’? I don’t understand if you are trying to say that high concentration of PAHs in topsoil layer also happen for dicofol and lindane? I cannot find a relationship between PAHs and those pesticides in this case. Please, explain this item to me.
Response: The sentence about dicofol and lindane was wrongly included. Now removed
· Line 264. According data from table 3, mean concentration for total PAH in paddy fields is 61.54 µg kg-1.
Response: corrected (line 284)
· Authors are using numbers to identify sample sites (S1-S18). Please, add the corresponding number when talk about them using their names (lines 286-289).
Response: Corresponding names added (line 307-310). All the corresponding names have mentioned in footnote of Figure 1.
· In Table 5 two diagnostic ratios are not right written: Ph/An (it should be Phe/Ant) and Flu/Py (Flu/Pyr). In this last case, I am not sure if it must be Flu or Flr. There is not accordance between the table and the text.
Response: changes made table 5 (page 19)
· Talking about Ecotoxicological and human Health Risk Assessment, in line 348, I think that the right mean value of RQ(MPCs) is 0.219 in the 0-10 cm layer.
Response: Comment included (line 376)
· Table 6. Please, to simplify and better understand, add a footnote explaining what is the meaning of every head of the columns.
Response: the table is now easily understandable and footnote added
· I think there is a mistake in the series of TEQ values in line 372-373. According results, I think that the right order is BaP (34.01) < BaA (0.351) > Chr (0.054) > Ant (0.049) > Ace (0.026) > Flr (0.020) > Pyr (0.015) > Phe (0.011) > Flu (0.008) > Acy (0.008) > Nap (0.004).
Response: comment considered. The results from the two soil depth presented separately (line 409-412).
· Please, check if the total TEQ of this study (line 376) is right. According data from table 6, I think it is 34.55. Please, verify it.
Response: The result for the 10-20 cm depth was missed. The corrected results are included (line 414). The total TEQ values of this study, i.e. 34.55 µg kg-1 in the top and 24.74 µg kg-1
· References 10, 12, 27, 28, and 36 are not found in the text. I have been unable to find them.
Response: The references were missed due to Mendeley desktop error. Now included
· All citations along the text should be placed in squared brackets instead of normal brackets.
Response: normal brackets “()” changed by squared brackets “[]”

Reviewer 2 Report
Thank you for the opportunity to review this manuscript. Comments are attached.

Author Response
Reviewer 2
Introduction
· The authors state that LMW PAHs are abundant and highly toxic, but are less persistent, low carcinogenic, and more easily degradable. The reference cited for this statement is not relevant and does not support the statement. Additional references such as Johnsen et al. 2005 Environmental Pollution – Principles of microbial PAH-degradation in soil, may be helpful to the authors in clarifying their statements on what is known about transport/fate of PAHs in soil, which will help them to better formulate explanations of their experimental results.
Response: Comment considered. I have also some supporting papers like: (Kumar et al, 2014). “Polycyclic aromatic hydrocarbon fingerprints in the Pichavaram mangrove-estuarine sediments, southeastern India’
Materials and Methods
· There is no explanation or discussion of the types of exposure expected in these areas. How do people regularly use these locations? There needs to be some justification of the methods used to assess exposure. Should children be considered?
Response: The district covers large area used for agricultural production. The dietary requirement for large portion of the population in the city of Wuhan and the study area depends on the crops, vegetable, and fruits produced from the district. (Line 77-80)
· There is no clear explanation of how DL and LQ shown in Table 2 were defined
Response: It has explained (Line 162-165)
· What types of blanks were collected in this study?
Response: There were no blank samples collected from the field, the term blank here is to mean (the solvent used in for extraction)
· Site numbers used in tables should also be listed on Figure 1
Response: Comment considered: Numbers included in Figure 1 (page 6)
· Many statements are made throughout the manuscript about certain groups of PAHs being found at greater concentrations than others, but rarely is any kind of significance reported, and in many cases the values are quite close. Statements about differences between groups of values should be followed with a measure of statistical significance in any case where they are important to the discussion.
Response: Significance value included in the manuscript (line 254-256). One-way analysis of variance (ANOVA) revealed a significant variation in concentration of individual PAHs (p < 0.05).
· There seems to be some confusion between whether 16 or 11 PAHs were measured. Different numbers are used in different sections. Clarify this and make sure it’s.
Response: the list of the 16 PAHs is the standards we purchased, but the actual number PAHs investigated were 11. These 11 PAHs are commonly reported at high concentration in china.
· Lines 218-227: The comparison to literature is confusing and redundant with Table 4. It does not appear from the numbers that midway atoll is significantly greater in concentration. The authors do not address differences in the number of PAHs measured anywhere in the text.
Response: The number of PAHs and values for midway atoll has included in (Table 4). (Page 16)
· Lines 229-230: This states that the same three PAHs were most abundant in both layers, so there is no significant comparison. Should be re-phrased. More generally, a comparison of composition visually (bar chart or similar visualization) would be very helpful to the reader to understand where there’s any difference in composition among site types.
Response: The sentence has rephrased. The three compounds were dominant on both soil depths (line 249-250).
· Line 244 requires a citation
Response: It was wrongly included, and it has removed
· Table 3: It is very hard to believe that there are any differences in concentration that are significant between top and sub-surface based on ranges shown here, but much of the discussion is based on this observation. Please provide a stdev for any averages shown, and an alpha or p-value as measure of significance of differences between groups of numbers. Otherwise, many conclusions in this section cannot really be drawn. Also, make sure significant figures in reported values are consistent.
Response: The statistical significance values for different land-use types has included in (line 284-286). No statistically significant difference inland-use types (p < 0.05).
· Lines 300-304: This is a very hurried explanation of potential correlations between individual PAHs, and no statement is made about why these correlations would matter. A table showing strength of correlation between all PAHs would be much more useful, even if it is in the Appendix.
Response: We have tried to look the relationship between soil properties and PAHs, because soil properties can alter the environmental availability of PAHs. As well as the relationship between PAHs is helpful to predict the compounds with similar source, so that it will help the researchers in identifying the potential sources. The detail correlation has displayed in appendix (Table 8) (Page 25)
· Line 348: Conclusions about the risk posed by PAHs in soil should be based on individually calculated RQs for different samples, not averages. It is not clear from the table whether there are any RQs above thresholds – it is not meaningful to just state the average is below. Also, the authors state that there is a lot we do not know about toxicology of PAHs, so the blanket statement that there is not risk posed at this site should be changed to communicate that there is still uncertainty.
Response: Comment considered. The individual and total RQ(MPCs), RQ(NCs) and TEQs for each compound has clearly presented in Table 6. (Page 21)
· There is little discussion of what sources are present besides petrogenic – are they distant or local? Might they differ between sites? Is there any spatial trend throughout the region? Are PAHs mostly introduced by atmospheric deposition of particulate or some other route? More clear details on these points are needed
Response: Some points added in the manuscript (line 371-373)
· The authors should justify how they chose the TEFs used. Other recent articles on this topic in IJERPH, including Pehnec & Jakovljevic 2018, consider multiple TEFs.
Response: Comment considered. These values are commonly used by researchers and accepted values.
· Lines 361-379: Again, mean values are not very meaningful. A range of values for different samples needs to be provided. ILCR results >10-6 can indicate potential risk, such as the value reported here. Consult recent article Shabbaj et al. 2018 IJERPH – they did a good job of summarizing risk metrics.
Response: as per the comments given here the conclusion about ILCR values were discussed based on the obtained range values (line 400-405). Moreover, the ILCR for children were included appendix (Table 9).

Reviewer 3 Report
This paper conducts a series of assessments of PAHs found in soil samples, which are sampled from Wuhan, China. The approaches and results are quite good, but the article needs a lot of editing work to help readers understand the content better, considering so many compounds and indices present in the paper.
Line 22, please indicate what PAH stand for here in the beginning.
Lines28-29, the authors may want to define 0-10 cm as top and 10-20 cm as subsequent layers.
Line 57, please check the definition for HMW. In Table 2, they are up to 5 rings.
Line 64, full name for POPs should be given.
Line 85, is it 12.3%, instead of 123%?
Line 142, with LOD being 0.008-0.168, LOQ should usually be higher than LOD. Please check the original data.
Lines 229-230, they are the same three PAHs in the top and subsequent layers, but in different orders. Please rephrase the sentence to make it clear.
Line 233, what are the reasons? You are not asking readers to find the reference to get the answer, are you? Please state the reasons here.
Lines 255-256, the abbreviations should be difined somewhere earlier in the text.
Lines 301-304, what do these strong correlations among PAHs indicate?
Line 326, TI is not defined earlier in the text.
Lines 328-336, the suthors should consider to place a short summary at the end of this paragraph.
Line 365, 3.58X10-6 is larger than the standard, 10-6. It is a bit of concern, and the authors should check the calculation or re-write the conclusion about it.
Author Response
Reviewer 3
· Line 22, please indicate what PAH stand for here in the beginning.
Response: Definition has added in the beginning (line 22)
· Lines28-29, the authors may want to define 0-10 cm as top and 10-20 cm as subsequent layers.
Response: Top and subsequent layers changed by 0-10 and10-20 cm (Line 30-31)
· Line 57; please check the definition for HMW. In Table 2, they are up to 5 rings.
Response: change in definition of HMW PAHs has done and explained in (line 60)
· Line 64, full name for POPs should be given.
Response: Full name added: persistent organic pollutants (POPs) (line 67)
· Line 85, is it 12.3%, instead of 123%?
Response: it was wrongly written and has corrected as 12.3% (line 93)
· Line 142, with LOD being 0.008-0.168, LOQ should usually be higher than LOD. Please check the original data
Response: we have checked the data: the range for LOQ was written wrongly and corrected as 0.025–0.560 µg kg-1 (line 166).
· Lines 229-230, they are the same three PAHs in the top and subsequent layers, but in different orders. Please rephrase the sentence to make it clear.
Response: The three compounds were dominant on both soil depths (line 249-250).
· Line 233, what are the reasons? You are not asking readers to find the reference to get the answer, are you? Please state the reasons here.
Response: Reasons mentioned and included in the text (line 253-254)
· Lines 255-256, the abbreviations should be defined somewhere earlier in the text.
Response: We have defined in the methodology part (line 96-97)
· Lines 301-304, what do these strong correlations among PAHs indicate?
Response: interpretation has given (line 325-326)
· Line 326, TI is not defined earlier in the text.
Response: we have defined it (line 349-351)
· Lines 328-336, the authors should consider placing a short summary at the end of this paragraph.
Response: Explanations and changes are written (line 353-354)
· Line 365, 3.58x10-6 is larger than the standard, 10-6. It is a bit of concern, and the authors should check the calculation or re-write the conclusion about it.
Response: Changes have done in the interpretations (line 400-404) & conclusion (line 435)
In addition to reviewers comments the following changes are also included
· In the footnote for equations 1-6, the semicolon used to separate replaced by comma (line 191-194)
· Table 1 a bold line added to separate individual and ∑PAHs (line 200)
· In the footnote of equations 7-10 semicolon used to separate replaced by comma (line 213-223)
· The numbers 1st, 2nd & 3rd are changed by first, secondly and lastly (line 346-348)
· Table 6 lines to separate columns were added (page 19) & to make the table clear “PAHs quality values” has added in the 1st row of the 2nd column
· Table 9 a column for sum ILCR and ILCR values for children added (page 25).
· Acknowledgement part “ Mr. Benjamin Muema Watuma” added. He helped us to edit the language aspect of the manuscript.

Round 2
Reviewer 1 Report
In this new revised version, authors have included many changes suggested by reviewers. However, in my opinion, English must still be revised because there are wrong concordances and missing words (verbs). For instance:
· Line 28. A gas chromatograph equipped…(besides, it can be written with small letters)
· Line 49. PAHs are often generated from…
· Line 66. Researchers have affirmed…
· Line 157. The calibration curve for compounds was…
· Line 158. Coefficient value (R2= 0.991) that implies…
· Line 263. … study area was evaluated…
· Line 341. It is also possible to apply a pyrene index (PI)…
I also think that capital letters must be used for ‘United States Environmental Protection Agency (USEPA)’ in line 57, or ‘Chinese Society of Soil Science’ in line 168.
1. Previous considerations
· I still am very surprised by the comment of the authors about; they wrongly wrote ECD instead of FID as the detection system of the gas chromatograph. In the same line, I would like to say that ‘medium’ is not a good word to talk about sample type. I consider that word is appropriated to talk about substances for growing cells. For instant, environment could be used for it, although in line 55 it could be a redundancy.
· I also think that it is necessary to add grade, company and country for HCl (line 131) or hexane (line 155) as for the furnace described in line 155.
· Please, in the same way as all the PAH described, change the capital letter for ‘Dibenzo[a,h]anthracene’ (line 118).
· Add a blank among ‘10mL’ in line 138, or in ‘Barren land(BL)’ in table 3.
· Footnote, line 222. Delete bold for ‘15 kg’ inside the parenthesis and unify font for ‘2800’ in SA.
· Change into bold letters the sigma symbols in Table 2 (ΣLMW) in the first column and in the footnote. Center the last value of the ‘Mean (0-10 cm)’ column.
· Line 310. Close quotation marks before Table 3.
· Please, unify font size for Pyr (0.169) in line 377. Unify too values for ΣLMW in Std (0-10 cm) and Std (10-20 cm) columns.
· What is the meaning of the symbol ‘&’ in Table 1? I think it must be deleted.
2. Results
· Although it is a tiny detail, if the result in Table 2 for mean value of ΣLMW is 73.02, please use the same for the comment in line 251.
· In line 308, there are five samples that can be grouped as ’not contaminated’.
· In line 320, authors say that Pearson’s correlation value obtained in this study showed a significant positive correlation among individual PAHs. However, according Table 8, the couple Nap/BaA is light negative (-0.053).
· In line 324, authors say that BaA showed strong positive correlation with Pyr and Flu. I think it is a mistake because they are Pyr (0.439) and Fla (0.505).
· In line 328, not only BaP (R= 0.191) shows a positive correlation, because Ant shows an R= 0.137.
· In the following lines, R-value for Ant is 0.139 (instead 0.138) and Flr is 0.022 (instead of 0.002).
· In line 387, I consider that seventh compound with RQNC value higher than 1 is not Flu (0.32) but Pyr (16.87).
3. References
· Please, follow the instructions for authors to adapt all references according to https://www.mdpi.com/journal/ijerph/instructions
Author Response
Reviewer 1
· Line 28. A gas chromatograph equipped…(besides, it can be written with small letters)
Response: Changes made appeared in (Line 27)
· Line 49. PAHs are often generated from…
Response: Changes made appeared in (Line 49)
· Line 66. Researchers have affirmed…
Response: Changes made appeared in (Line 55)
· Line 157. The calibration curve for compounds was…
Response: Changes made appeared in (Line 159)
· Line 158. Coefficient value (R2= 0.991) that implies…
Response: Changes made appeared in (Line 160)
· Line 263. … study area was evaluated…
Response: Changes made appeared in (Line 265)
· Line 341. It is also possible to apply a pyrene index (PI)…
Response: The sentence has rephrased and appeared in (Line 344-345)
· I also think that capital letters must be used for ‘United States Environmental Protection Agency (USEPA)’ in line 57, or ‘Chinese Society of Soil Science’ in line 168.
Response: Changes made appeared in (Line 56 & 170)
· I still am very surprised by the comment of the authors about; they wrongly wrote ECD instead of FID as the detection system of the gas chromatograph. In the same line, I would like to say that ‘medium’ is not a good word to talk about sample type. I consider that word is appropriated to talk about substances for growing cells. For instant, environment could be used for it, although in line 55 it could be a redundancy.
Response: The word mediums has changed by “Environmental components)
· I also think that it is necessary to add grade, company and country for HCl (line 131) or hexane (line 155) as for the furnace described in line 155.
Response: The company name included for HCl and Hexane “Sinopharm Chemical Reagent Co., Ltd., China” (Line 130 & 157) and Muffle furnace ‘Zhengzhou Protech Furnace Co., Ltd., China” (Line 156)
· Please, in the same way as all the PAH described, change the capital letter for ‘Dibenzo[a,h]anthracene’ (line 118).
· Add a blank among ‘10mL’ in line 138, or in ‘Barren land(BL)’ in table 3.
Response: Added (Line 138) and Table 3 (Page 14)
· Footnote, line 222. Delete bold for ‘15 kg’ inside the parenthesis and unify font for ‘2800’ in SA.
Response: Font style normalized (Line 224)
· Change into bold letters the sigma symbols in Table 2 (ΣLMW) in the first column and in the footnote. Center the last value of the ‘Mean (0-10 cm)’ column.
Response: Changes made (Page 13)
· Line 310. Close quotation marks before Table 3.
Response: Quotation included ( Line 313)
· Please, unify font size for Pyr (0.169) in line 377. Unify too values for ΣLMW in Std (0-10 cm) and Std (10-20 cm) columns.
Response: Unified
· What is the meaning of the symbol ‘&’ in Table 1? I think it must be deleted.
Response: It is supposed to express 1 < values < 800. & change by semicolon “;”
· In line 308, there are five samples that can be grouped as ’not contaminated’.
Response: Three in the 0-10 cm and four in the 10-20 cm
· In line 320, authors say that Pearson’s correlation value obtained in this study showed a significant positive correlation among individual PAHs. However, according Table 8, the couple Nap/BaA is light negative (-0.053)
Response: Changes made
· In line 324, authors say that BaA showed strong positive correlation with Pyr and Flu. I think it is a mistake because they are Pyr (0.439) and Fla (0.505)
Response: It has rephrased (strong changed by Significant) (Line 325-326)
· In line 328, not only BaP (R= 0.191) shows a positive correlation, because Ant shows an R= 0.137
Response: Included
· In the following lines, R-value for Ant is 0.139 (instead 0.138) and Flr is 0.022 (instead of 0.002).\
Response: Corrected (Line 333-334)
· In line 387, I consider that seventh compound with RQNC value higher than 1 is not Flu (0.32) but Pyr (16.87).
Response: Corrected (Line 388)
· Please, follow the instructions for authors to adapt all references according to https://www.mdpi.com/journal/ijerph/instructions
Response: Reference adjusted as per the guide
Reviewer 2 Report
Thank you for the opportunity to review this revised manuscript. At this time, the authors have addressed my major comments from the first review and I have no further major comments.
I found that there are areas where the grammar still needs to be improved and the authors should be careful to define acronyms the first time they are used, but beyond this, I recommend the manuscript be accepted for publication.
Author Response
Reviewers 2
· I found that there are areas where the grammar still needs to be improved and the authors should be careful to define acronyms the first time they are used, but beyond this, I recommend the manuscript be accepted for publication.
Response: Thank you for your comments we have thoroughly checked the grammar and English problems
Reviewer 3 Report
The revision has been made accordingly. I have no further question or comment.
Author Response
Reviewer 3
· The revision has been made accordingly. I have no further question or comment.
Thank you for your comments we have thoroughly checked the grammar and English problems